# Vortex Dynamics in the Sinus of Valsalva

**DOI:** 10.3390/bioengineering12030279

**Published:** 2025-03-11

**Authors:** Jiaxuan Fan, Elias Sundström

**Affiliations:** Department of Engineering Mechanics, FLOW Research Center, KTH Royal Institute of Technology, 100 44 Stockholm, Sweden; elias@kth.se

**Keywords:** aortic heart valve, fluid-structure interaction, sinus vortex dynamics

## Abstract

Patients undergoing aortic valve repair or replacement with associated alterations in stiffness characteristics often develop abnormalities in the aortic sinus vortex, which may impact aortic valve function. The correlation between altered aortic sinus vortex and aortic valve function remains poorly understood due to the complex fluid dynamics in the aortic valve and the challenges in simulating these conditions. The opening and closure mechanism of the aortic valve is studied using fluid–structure interaction (FSI) simulations, incorporating an idealized aortic valve model. The FSI approach models both the interaction between the fluid flow and the valve’s leaflets and the dynamic response of the leaflets during pulsatile flow conditions. Differences in the hemodynamic and vortex dynamic behaviors of aortic valve leaflets with varying stiffness are analyzed. The results reveal that, during the systolic phase, the formation of the sinus vortex is closely coupled with the jet emanating from the aortic valve and the fluttering motion of the leaflets. As leaflet stiffness increases, the peak vorticity of the sinus vortex increases, and the phase space of the vortex core develops a pronounced spiral trajectory. During the diffusion phase, the vortex strength decays exponentially, and the diffusion time is longer for stiffer leaflets, indicating a longer residence time of the sinus vortex that reduces the pressure difference on the leaflet during valve closure. Changes in leaflet stiffness play a critical role in the formation and development of sinus vortices. Furthermore, the dynamic characteristics of vortices directly affect the pressure balance on both sides of the valve leaflets. This pressure difference not only determines the opening and closing processes of the valve but also significantly influences the stability and efficiency of these actions.

## 1. Introduction

As the heart pumps blood around the body, heart valves are crucial in converting the heart’s contracting chambers into an effective pump by ensuring one-way blood flow. The aortic sinus, also known as the sinus of Valsalva, consists of three half-moon-shaped dilated areas at the root of the aorta. The complex biological structure of the aortic sinus and valves enables them to withstand billions of passive deformations over a lifetime, demonstrating incredible durability and efficiency [1,2,3]. Since Leonardo da Vinci first described the aortic sinus [4], a series of subsequent studies have revealed the specific role in influencing blood flow and the valve opening and closing dynamics. It is now well accepted that the vortical flow within the aortic sinuses plays a role in stabilizing the valve opening and reducing pressure fluctuations by regulating the pressure relationship between the sinus ridge and the sinus cavity, both at the moment when the valve is fully open and during its gradual closure [5]. Many studies have also explored the effects of aortic sinus vortices on aortic valve dynamics and the onset and progression of calcifying aortic valve disease [6,7,8]. A key flow structure observed is the counter-rotating vortex at the upper end of the sinuses near the base of the lobules, which can greatly affect the directivity and strength of the shear stress along the base of the leaflet [9,10,11]. Recent in vitro studies have shown more complex hydrodynamic behavior, revealing the case for secondary vortices and instability mechanisms during laminar–turbulent transition [8,12]. In recent numerical studies of the aortic valve, the Kelvin–Helmholtz instability was found to have a much more significant effect on the valve dynamics as compared to vortices in the aortic sinus [13]. The various interpretations of fluid dynamics within the aortic sinus in the literature are often contradictory, indicating that the mechanism of the physiological flow conditions facilitated by the aortic sinus is not fully understood [7].

The opening and closing of the heart’s valves involve a strong interaction between the blood and surrounding structures. The actual motion is determined by the complex interaction of fluid dynamics and structural system properties, resulting in a coupled, multi-scale, and multi-physics fluid–structure interaction (FSI) problem [14,15]. The use of finite element analysis (FEA) to evaluate the performance of both native and artificial heart valves has been discussed by many studies in recent years, and FSI is commonly recognized to accurately capture the coupling between the aortic valve leaflets and the blood [16,17]. In FSI simulations, the fluid phase is most conveniently described by the Eulerian reference frame, while the Lagrange formulation is more suitable for the solid phase; however, these formulations are incompatible [18]. The Arbitrary Lagrange–Euler (ALE) method is widely used in arterial flow modeling. ALE requires constant updating of the fluid mesh to adapt to the movement of the domain [19,20,21]. For problems regarding large deformation or strong non-linear deformation, frequent mesh reconstruction or remapping may significantly increase computational complexity. In contrast, virtual domain formulation [22,23] circumvents updating fluid grids: fluids are described using fixed grids in the Eulerian reference frame, and Lagrangian multipliers handle the coupling of fluids and structures. Another widely used method for simulating the moving boundaries of the human heart and heart valves is the Immersion Boundary (IB) method [24,25]. The principle of the IB method is to treat the elastic material as part of a fluid that exerts an additional force. A frequent challenge in applying Peskin’s method is numerical stiffness, which necessitates using small time steps. Potential sources of the observed stiffness include the singular nature of the force field exerted by the IB on the fluid, as well as the inherent physical stiffness of the IB method itself [26]. The overset domain method allows for interpolation calculations between stationary background grids and component grids with overset grids, interpolating between donor cells and recipient cells for updating the geometry and grid due to the motion of the leaflets [27]. Compared with the ALE and IB methods, the overset domain method reduces the dependence on mesh deformation while providing the same accuracy and avoids producing low-quality elements in gaps of boundaries in close proximity, with improved solver stability and convergence characteristics [28]. The overset domain has been widely verified in various three-dimensional steady and unsteady laminar flows, showing strong agreement with the experimental data and simulation results [29].

Leaflet stiffening and its effect on the sinus vortex play a crucial role in the progression of aortic valve stenosis and calcific aortic valve disease (CAVD) [30]. Fluid shear stresses, alongside genetic factors and inflammatory pathways, have emerged as key regulators in the initiation and progression of CAVD. Hemodynamics within the sinuses of Valsalva are critical in influencing CAVD progression, although it remains unclear whether they initiate or exacerbate the disease. However, studies have shown a strong correlation between hemodynamics and disease [31]. CAVD impairs normal valve function, leading to restricted valve opening, higher jet velocities, larger sinus vortex space, and non-physiological flow, with low shear stress in the sinus being a significant indicator of the disease [32].

This study aims to reveal and analyze the hydrodynamic characteristics of the aortic valve and the local flow characteristics within the sinus during the systolic and diastolic phases of the heart. A specific emphasis is placed on the investigation of how the sinus vortices are formed and maintained during valve opening and closing, and their impact on valve function. The FSI simulations based on an idealized two-dimensional model of the aortic valve were compared with the experimental results and the inviscid flow solution to Hill’s spherical vortex mode obtained by Bellhouse and Talbot [5]. The main research objective is to evaluate the dynamic influence of the aortic sinus vortex during valve opening and closing and to understand the mechanism of action for effective aortic valve function.

## 2. Material and Methods

### 2.1. Governing Equations and Numerical Method

The interaction between the aortic valve and blood flow was simulated using a 2-way coupled FSI solver. The FSI simulations were performed with an implicit unsteady scheme where the fluid was discretized with a second-order scheme. The governing equations for fluid flow are the incompressible Navier–Stokes continuity and momentum equations [8]. They are given as∇·u=0$$\frac{\partial u}{\partial t}+u\cdot \nabla u=-\frac{1}{\rho }\nabla p+\nu \nabla^{2}u+f$$
where u represents the fluid velocity field, *p* is the fluid pressure field, and ρ and ν are the fluid density and dynamic viscosity, respectively. f is the external body force. The deformation of the solid leaflet is described by an equation for the conservation of linear momentum$$\rho_{s}\frac{\partial d_{s}}{\partial t}+\rho_{s}\left(d_{s}\cdot \nabla \right)d_{s}=\nabla \cdot \sigma_{s}+f_{s}$$
where $\rho_{s}$, $d_{s}$, $\sigma_{s}$, and $f_{s}$ are the solid density, displacement, stress tensor, and body force per unit volume, respectively. The hyperelastic neo-Hookean model is used to take into account non-linear stress–strain behavior. The incompressible (volume ratio J = 1) neo-Hookean strain-energy density function is$$W=C_{10}\left(I_{1}-3\right)$$

Parameter $C_{10}$ is a material constant related to the shear modulus, representing the stiffness of the material under shear deformation. $I_{1}$ is the first invariant of the right Cauchy–Green deformation tensor $C=F^{T}F$. The deformation gradient tensor F, $F=\nabla u_{s}+I$, describes the change in the position of material points relative to the initial configuration.

The interaction of fluid and structure motion is implemented with boundary conditions at the interface between the structure and the fluid. The following formulas represent displacement continuity, normal velocity continuity, and stress equilibrium at the interface, respectively.$$d_{\Gamma }^{f}=d_{\Gamma }^{s}$$$$\hat{n}\cdot u=\hat{n}\cdot \frac{\partial d_{\Gamma }^{s}}{\partial t}$$$$PF^{T}n=\sigma_{f}n$$In the equations above, $d_{\Gamma }^{f}$ and $d_{\Gamma }^{s}$ represent the displacement vectors of the fluid and solid. P is the first Piola–Kirchhoff stress tensor, representing the internal stress of the solid, whereas $\sigma_{f}$ is the Cauchy stress tensor of the fluid, and n specifies the normal vector at the interface, used to describe the direction of velocity and stress.

The SIMPLE (Semi-Implicit Method for Pressure Linked Equations) algorithm, implemented in the software STAR-CCM+ version 2406 was used to solve the fluid flow, and the calculation of each node displacement is conducted using B-splines [33]. Details of this solver and its mathematical validation can be found in [34].

The solid mesh is composed of hexahedron elements, e.g., Hex8, while the fluid grid is discretized with polygonal elements and with prismatic cells adjacent to no-slip walls. An overset region with the least squares interpolation scheme is considered to enable large leaflet deformation, which maintains computational efficiency and accuracy. Hole cutting with zero-gap treatment is adopted in conjunction with the overset algorithm to enable complete closure of the aortic valve during diastole.

Adaptive time stepping was used to improve the quality and run time of the simulation due to the large variations in flow topology and varying time scales of the physics. The initial time step was a millionth of the cardiac cycle. During the simulation, the maximum norm of the leaflet displacement change was monitored. This was used to automatically regulate the time step change between a minimum factor of 0.25 and a maximum factor of 1.5.

### 2.2. Two-Dimensional Parametrized Aortic Model

Due to the complex 3D geometry of the aortic valve, the actual aortic root with three relatively symmetric sinuses and corresponding leaflets is often idealized into a two-dimensional model [13]. A longitudinal section is considered that cuts the midpoint of the leaflet (see Figure 1). The geometry of the aortic wall was based on data from Bellhouse et al. (1969) [5], and geometrical parameters are listed in Table 1, which are constant throughout each analyzed case.

The aorta wall was modeled as a rigid body, and the leaflet material was modeled as a non-linear hyperelastic neo-Hookean material with a density equal to 1120 kg/m^3^, Young’s modulus equal to 2 MPa, and a Poisson ratio equal to 0.49. The fluid domain is treated as a Newtonian incompressible fluid with a density of 1050 kg/m^3^ and a dynamic viscosity of 0.0035 Pa·s.

The solid boundaries surrounding the fluid inlet and outlets were fixed. The inlet condition was set on the fluid domain as the pressure difference between the aorta and left ventricle (see Figure 2), and the pressure of the fluid exit was zero gauge. The pressure difference calculation was based on the CircAdapt lumped-parameter model of the heart and circulatory system [35].

Four cardiac cycles were simulated to minimize the influence of initial transient behavior on the solution (the maximum difference of leaflet stress in 4 cycles is reduced to less than 3%). Each cycle lasted 0.85 s, with the opening and closing process accounting for 32% of the cycle, or about 0.27 s, and the valve was completely closed the rest of the time. All results presented in this research are taken from the fourth cardiac cycle. The analysis of the grid sensitivity on the results is presented in Appendix A.

## 3. Results

### 3.1. Vortex Behavior in Steady Flow

The qualitative comparison between the simulation results and the distribution of the pressure coefficient obtained from Hill’s spherical vortex in steady flow with velocity u=0.06 m/s is shown in Figure 3 [5].

Near the primary vortex within the sinus, both results exhibit a similar pressure distribution, with the minimum pressure appearing at the stagnation point within the vortex center, while the high-pressure region is primarily located along the sinus edge, far from the vortex center. These points occur at $r=r_{a},\theta =0,\pi$. In the simulation, the primary vortex center shifts slightly towards the cusp and the sinus gap, resulting in asymmetry in the pressure distribution. This contrasts with Hill’s spherical vortex model, which predicts a symmetric pressure distribution with the vortex located at $\theta =90^{\circ }$ and a radial distance of $r_{a}/\sqrt{2}$. The asymmetry observed in the simulation arises from the more complex inflow and outflow conditions across the sinus gap that are not accounted for in the potential solution of Hill’s spherical vortex model. There are also significant viscous effects and variations in pressure gradients around the sinus and leaflet boundaries, leading to the displacement of the vortex core and vortex deformation that are unaccounted for in Hill’s spherical vortex model.

The simulation result also shows that the formation of local small vortices near the leaflet root leads to deviations in the local pressure distribution. This can be considered a result of local recirculation and overlapping vortex structures generated under complex boundary conditions. However, such small-scale flow features are not accounted for in Hill’s spherical vortex model.

### 3.2. Effect of Leaflet Stiffness in Pulsatile Flow

For the variation in the valve opening area over time, a comparison was conducted between the simulation results and the experimental results previously obtained by Bellhouse et al. [5]; see Figure 4a. Throughout the systolic phase, the characteristics of a rapid valve-opening phase and a slower closing phase are exhibited. A higher stiffness yields a faster opening as compared to a lower stiffness, which reflects that a softer leaflet can respond more sensitively to pressure changes in the dynamic flow field.

The difference between the numerical results and the experimental results lies in the dynamic behavior of the valve during its opening state. The FSI simulation reveals valve flutter, transitioning from strong oscillations to a stabilized state. The decay of the oscillation is the combined effect of viscous damping and the leaflet’s stiffness damping. The overall damping ratio for the case E=2 MPa is $\xi =\delta /\sqrt{\delta^{2}+{\left(2\pi \right)}^{2}}=1.5\%$. The associated stiffness damping is then given by $\alpha =\xi /\left(\pi *f\right)=1/\pi \cdot 10^{-4}$ s, where the frequency of the damping response is f=14 Hz. These levels of the damping response are of similar order to other FSI models [8,36]. Another notable difference is the gradual reduction in the opening area during the midsystolic phase in the FSI simulation. This is caused by the decreasing pressure drop between the aorta and the left ventricle obtained with the lumped-parameter model (see Figure 2), and this has also been observed in other FSI studies [37,38].

Figure 4b illustrates the pressure difference variation during the valve closure. A higher stiffness results in a delayed closure that correlates with a delayed peak pressure difference. The difference in timing of the peak pressure difference between the simulation and experimental results can be attributed to differences in geometry and the applied boundary conditions.

The influence of leaflet stiffness on the aortic flow and flow into the sinus cavity is presented in Figure 5. The simulated results show a consistent trend with the experiment, indicating that the flow rate is mainly affected by the pressure difference imposed on the inlet and outlet of the control volume (see Figure 2).

The flow across the sinus gap into the sinus during the closing of the aortic valve is substantial until the point of closure. Bellhouse hypothesized that, once the valve closes, the established flow pattern during early systole would persist. However, the simulated results show a sudden change during closure where the net flow rate will show a damped oscillation and eventually return to zero before the next systole. It should be noted that Bellhouse calculated the flow into the cusp–sinus cavity from the rate of displacement of fluid by the leaflets, whereas, in the simulation, it is integrated across the sinus gap. The flow across the sinus gap into the sinus during the closing of the aortic valve also shows a significant dependence on the leaflet stiffness, where a lower stiffness results in an earlier closure.

Figure 5c presents the stress–strain relationship for different cases of leaflet stiffness. In the low-strain range (strain < 0.05), the stress–strain curve shows an approximately linear relationship, reflecting typical elastic behavior. At higher strains, the strain-hardening effect of the material under large deformation becomes increasingly significant. The inflection point (strain ≈ 0.12) corresponds to the dynamic transitions between valve opening and closing, while the later flattening of the curve reflects the force situation when the valve reaches a stable state (either fully open or closed).

### 3.3. Unsteady Vortex Solution

Different vortex structures are generated in the aorta and the aortic sinus due to the moving leaflet. These vortices are marked in Figure 6. During the early phase of the systolic valve opening, there is evidence favoring the hypothesis that the sinus vortex is coupled with the jet flow emanating from the aortic valve (Figure 6a,b). The velocity near the leaflet edge shows expanding streamlines and the emergence of a starting vortex that is separating from the higher vorticity boundary layer on the aortic side of the leaflet. In the wake of the separated vortex, there is a growing shear layer separating the higher jet velocity from the nearly stagnant flow in the adjacent sinuses. At the sinus ridge, the flow splits, with one part progressing downstream in the aorta and the other being convected into the sinus and beginning to develop a main sinus vortex (Figure 6c,d). As time evolves to peak systole (Figure 6e), the sinus vortex displaces radially further into the sinus and is no longer fed with mean flow from the jet. This marks the end of the convective period and defines the beginning of the diffusion period of the sinus vortex. Towards the closing of the aortic valve (Figure 6f), the sinus vortex fills most of the sinus, and the relatively low velocity indicates that viscosity causes a reduction in vortex strength.

The trajectory of the vortex core in the sinus is shown in Figure 7a. During the convection phase, the path of the vortex core is restricted by the sinus wall, and the trajectory follows the curvature of the viscous non-slip wall, forming an initial single-vortex structure. With the accumulation of shear stress in the boundary layer, the fluid near the wall gradually develops convective instability. In the region downstream of the main vortex path, the local pressure gradient changes due to the combined action of viscous dissipation and reversed flow, which triggers the formation of a secondary vortex (marked by a red arrow) in the opposite direction near the wall, resulting in changes in the strength of the main vortex and its trajectory.

The trajectory of the sinus vortex during the diffusion phase was least-squares fitted to a logarithmic spiral using MATLAB version R2024a (The MathWorks, Natick, MA, USA):$$r=ae^{k\phi }$$
where *a* is the initial radius, which determines the starting size of the spiral, and *k* is related to the pitch angle α by k=tan(α). The initial radius values *a* obtained for E=1,2, and 3MPa were 40,31, and 26, respectively, while the *k* values were −0.80,−0.57, and −0.46, respectively. The curvature was computed using the formula $\kappa =\frac{1}{r\sqrt{1+k^{2}}}$, and the arc length was calculated with $L\left(\phi_{1},\phi_{2}\right)=\frac{\sqrt{1+k^{2}}}{k}\left[r\left(\phi_{2}\right)-r\left(\phi_{1}\right)\right]$. The results for mean curvature were $\kappa_{E=1}=0.056$, $\kappa_{E=2}=0.062$, and $\kappa_{E=3}=0.065$, while the arc lengths were $L_{E=1}=3.9$, $L_{E=2}=5.3$, and $L_{E=3}=6.5$.

As the leaflet stiffness increases, the aspect ratio of the spiral trajectory increases. Low-stiffness leaflets tend to deform more easily with the flow, resulting in a larger opening area (see Figure 5). The vortex core during the convection phase is initiated closer to the sinus ridge (Figure 7a). In contrast, with higher stiffness, the vortex is initiated at a larger distance from the sinus ridge, which correlates with increased radial displacement into the sinus cavity, which increases the curvature characteristics of the spiral trajectory.

Figure 7b shows the temporal evolution of vorticity of the sinus vortex and compares the conditions of different leaflet stiffness. During the convection phase, vorticity increases rapidly and reaches its peak, with higher leaflet stiffness generating a higher vorticity and a shorter convective time scale. A softer leaflet results in a longer convection time scale, characterized by time delay and a reduced peak vorticity.

The convective phase is a combined inflow and outflow mechanism that facilitates the establishment of circulatory motion within the sinus. The time required to generate the vortex within the sinus through convective processes, $t_{c}$, has been hypothesized—assuming potential flow—to be the time needed for a fluid particle to complete one circuit around the sinus cavity with a radius *a* and a peak aortic velocity of *U* [5]:$$t_{c}∼\frac{(2+\pi )a}{U}.$$Assuming a sinus radius of a=1.0cm and a peak aortic velocity of U=100cm/s, the characteristic time is approximately $t_{c}∼0.05\;s$. The simulated results for cases with different leaflet stiffness E=1, 2, and 3MPa yield convective times of vorticity growth of 0.034s, 0.013s, and 0.021s, respectively. The simulated result is of the same order of magnitude as the potential result and shows a trend of reduced convective time scale with increasing leaflet stiffness.

During the diffusion phase, the vortex strength decays exponentially, and a bi-exponential function $A\left(t\right)=A_{1}e^{-\lambda_{1}t}+A_{2}e^{-\lambda_{2}t}$ was used to fit the spiral path of the vortex core trajectory. The term with slower diffusion corresponds to a decay time $\tau_{D}=\frac{1}{\lambda }$, which corresponds to the ratio between the square of the characteristic diffusion length scale *a* over ν, the kinematic viscosity [5]:$$t_{D}∼\frac{a^{2}}{\nu }$$The decay times obtained for cases of different leaflet stiffness E=1MPa, E=2MPa, and E=3MPa are approximately 4 s, 8 s, and 12 s. It can be observed that, as the leaflet stiffness increases, the vortex diffusion process slows down, indicating that vortices associated with higher stiffness maintain their vorticity for a longer time. Overall, leaflet stiffness significantly affects the vorticity and time distribution across the convection and diffusion phases, with higher-stiffness leaflets generating increased residence time and peak vorticity.

### 3.4. Pressure Coefficient

The pressure coefficient plays a crucial role in understanding the net forces acting on the aortic valve leaflet throughout its dynamic motion. Figure 8 shows the forces that act upon the leaflet as a function of the pressure distribution from the root to the free edge. The force is expressed in non-dimensional form:$$C_{N}=\frac{1}{c}\left[\int_{0}^{c}\left(c_{p,p}-c_{p,s}\right)dx+\int_{0}^{c}\left(c_{f,p}\frac{dy_{s}}{dx}+c_{f,s}\frac{dy_{p}}{dx}\right)dx\right]$$This relation of the normal force coefficient is valid for a leaflet of arbitrary shape. The integral of the shear force is an order of magnitude smaller than the integral of the pressure force. Therefore, the net normal force acting on the leaflet during the cardiac cycle is mainly driven by the difference in pressure coefficient between the pressure and suction sides of the leaflet $\Delta c_{p}=c_{p,p}-c_{p,s}$. During the valve opening, the pressure differential across the leaflet contributes to its movement, where there is a larger difference starting at the root (Figure 8a). At the leaflet tip, the pressure must be equal on the pressure and suction sides, i.e., $\Delta c_{p}=0$ in that point, which is called the Kutta condition. This means that the integrand is =0 at the leaflet tip.

The reason for $\Delta c_{p}$ not being identical to 0 at the free edge in Figure 8 is that the points on the pressure and suction sides at the free edge are separated with a finite distance corresponding to the leaflet tip thickness.

In addition to the opening phase, Figure 8b,c shows the pressure difference distribution between the two sides of the leaflets with varying stiffness during peak systole and towards the phase of valve closure. During the opening phase, a net force acts to push the leaflets open (Figure 8a). Leaflets with higher stiffness exhibit reduced aortic opening area, leading to a faster convection time scale of the jet and resulting in a larger pressure difference and increased stress levels near the leaflet root (see Figure 5c). Once fully opened, the local extreme values of the pressure difference are closely associated with the position of the vortex core, as shown in Figure 8b. Higher vorticity contributes to a more symmetric pressure distribution on leaflets by balancing the pressure gradient, thereby effectively supporting the stability of the leaflets. During the closing phase, higher vorticity from the sinus vortex helps to reduce the pressure in the sinus and thereby also lower the net force on the closing leaflets. This correlates with a smoother and more controlled closure process.

**Figure 8 bioengineering-12-00279-f008:**
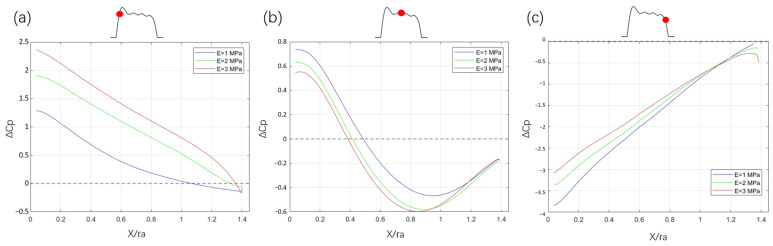
Pressure coefficient difference along the normalized valve length, calculated as suction side minus pressure side during the (**a**) opening, (**b**) fluttering, and (**c**) closure stages. The small inset plot (depicted with the black curve) illustrates the aortic opening area during systole, with the red dot marking the instantaneous time.

## 4. Discussion

In this study, differences in the hemodynamic and vortex dynamic behaviors of aortic valve leaflets with varying stiffness were analyzed. The results revealed that, during systole, the formation of the sinus vortex is closely coupled with the jet emanating from the aortic valve and the fluttering motion of the leaflets. As leaflet stiffness decreases, the expansion of the sinus vortex is restricted, and the influence of the sinus wall on the vortex core becomes more pronounced. During the diffusion phase, the vortex strength decays exponentially, and the diffusion time is significantly extended for stiffer leaflets, indicating greater stability and persistence of the vortex structure. These insights provide important biomechanical guidance for optimizing artificial valve designs. However, it was also observed that increased leaflet stiffness is accompanied by higher normal stress on the valve during systole, particularly at the regions of the valve root. Further research is needed to determine whether this increase in stress may heighten the risk of leaflet damage.

Due to the high spatiotemporal resolution of FSI simulations, we were able to reveal fine-scale hemodynamic features within the aortic sinus. Certain patterns, such as the growth of sinus vortices until late systole and the trajectory of the vortex core, were observed both in this study and in previous research [9,39]. This study addresses the challenge in identifying the origin of the main sinus vortex by adopting a bottom-up approach that breaks down complex flow problems into more isolated and focused investigations on vortex dynamics in sinuses. Based on the experimental work of Bellhouse et al. [5], it was suggested that excessively stiff valve leaflets perform inadequately; in such cases, if the valve cannot fully open, vortices fail to form, resulting in sudden and uneven behavior. However, our results indicate that the generation and development of main vortices within the sinus are not directly dependent on the valve opening area but are instead driven by local flow characteristics and fluid dynamic interactions. These discrepancies may stem from differences in research methods and technological approaches.

The primary advantage of coupled FSI simulation lies in its ability to analyze leaflet deformation and the effects of arterial tissue movement induced by hemodynamic variables. In the present study, we incorporated precise leaflet opening and closure dynamically, ensuring the stability of the results over several cardiac cycles. However, FSI analysis involves several assumptions. The first limitation is the assumption of rigid aortic and sinus walls. While these structures are inherently compliant, their motion is minimal compared to highly dynamic leaflets and thus is not expected to significantly influence sinus hemodynamics. Rosakis et al. suggested that a compliant aorta model may alter the distribution of wall shear stress (WSS) and wall-normal stress along the wall as the vessel deforms under these stresses [40]. Additionally, Hsu et al. found that the main differences between rigid and elastic models occur during valve closure. The compliant aorta expands [41] to absorb the fluid hammer impact and dissipate the initial kinetic energy into the surrounding tissues and interstitial fluids, thereby exhibiting lower flow rates and reduced valve oscillations [42]. The fluid hammer effect in our study using the rigid wall model may be overestimated [43].

In previous numerical studies on the hemodynamics of mechanical biological heart valves (BMHV) [44,45,46,47], it was commonly assumed that blood behaves as a Newtonian fluid given that the shear rate in large arteries is high and the blood viscosity reaches a constant value (3.0–4.0 mPa·s) [48]. Pawlikowski et al. and De Vita et al. compared a Newtonian model with the non-Newtonian Carreau model in terms of flow patterns and blood damage prediction. The results indicated that the valve dynamics, transvalvular pressure drop, and large-scale flow characteristics were similar between the two models, with the differences focusing on surface and time-averaged clinical parameters, such as WSS, pressure recovery, and blood damage [49,50]. However, Yeh et al. found that the non-Newtonian fluid model predicts slightly lower valve angular velocities and exhibits asynchronous motion during the closing phase [51]. Furthermore Yeh et al. showed that the shear-thinning behavior of the fluid influences the size of vortices near the valve and promotes the formation of secondary asymmetric flows [51]. Chuanhan et al. observed a mushroom-shaped turbulent structure appearing downstream of the valve leaflets and gradually diminishing as the valve Reynolds number increased [52]. Further research is needed for a physics-based understanding of how sinus vortices are influenced by non-Newtonian rheology models.

The isotropic hyperelastic neo-Hookean model for the leaflets in the present study did not account for histological effects, but it is commonly used for younger tissue specimens that have approximately isotropic characteristics. However, there is evidence that non-linearity and anisotropy increase with older aortas, suggesting the need for a more complex constitutive model formulation. The classic two-fiber-family Holzapfel–Gasser–Ogden (HGO) model, for example, has demonstrated better agreement for both younger and older tissue specimens [53]. Liogky et al. investigated how anisotropy in the constitutive model may affect leaflet coaptation. Their results showed that anisotropy had an insignificant effect on the coaptation characteristics but could influence the leaflet dynamics, resulting in smaller displacements [54]. Li et al. compared anisotropic and isotropic valves under the same loading conditions. It was found that the maximum longitudinal normal stress was larger in the anisotropic model, but the maximum transversal normal stress and in-plane shear stress were smaller [55].

Due to the limitations in our model, further improvements are suggested. Blood can be modeled as a non-Newtonian fluid, with viscosity varying according to shear conditions. The aortic root and sinus can be modeled as deformable structures with elastic material properties. Leaflets can be modeled as multilayer structures with aging characteristics and anisotropic material properties. These aspects are related to our future work.

## 5. Conclusions

In this study, we investigated the impact of leaflet stiffness on the vortex dynamics and flow stability within the aortic sinus. The findings revealed that changes in leaflet stiffness play a critical role in the formation, development, and pressure distribution of sinus vortices. Furthermore, we examined the influence of vortex dynamics on the pressure distribution across the leaflets, analyzing the regulatory effect on valve opening and closing behavior. The results indicate that the dynamic characteristics of vortices directly affect the transvalvular pressure gradient, which in turn significantly influences the stability and efficiency of the opening and closing processes. This study highlights the complex interplay between leaflet stiffness and vortex dynamics, offering guidance for future research and clinical applications aimed at optimizing the design of artificial valves.

## Figures and Tables

**Figure 1 bioengineering-12-00279-f001:**
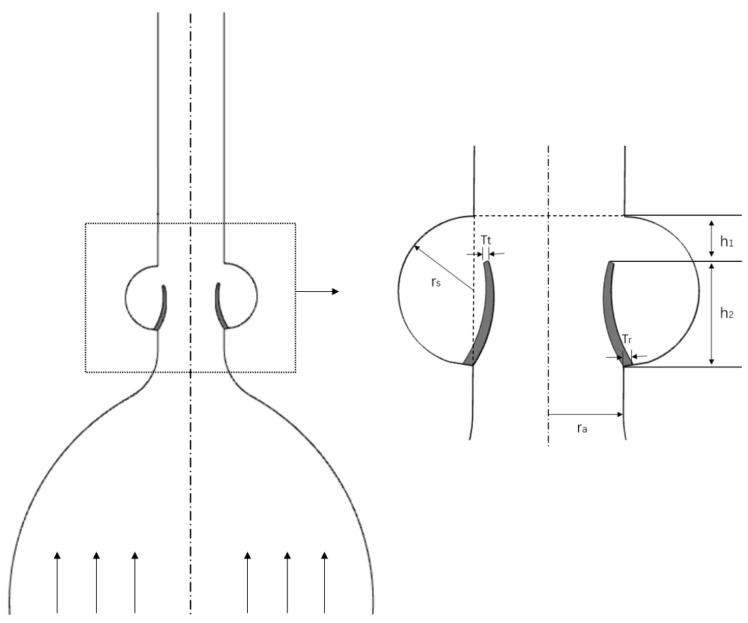
Parametrized two-dimensional model of the aortic wall with the aortic valve. The aortic sinus and leaflets within the black box are shown in a zoomed-in depiction to the right, with annotated dimensions for clarity.

**Figure 2 bioengineering-12-00279-f002:**
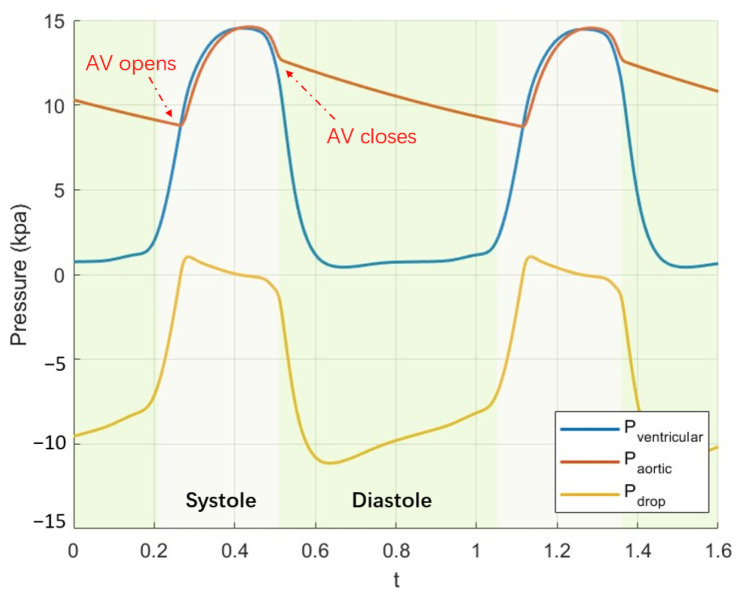
Transvalvular pressure gradient is defined as the difference in pressure between the left ventricle and the aortic outflow tract.

**Figure 3 bioengineering-12-00279-f003:**
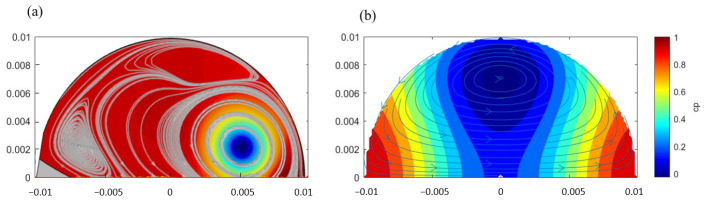
(**a**) Pressure coefficient distribution in the sinus of Valsalva in steady flow and (**b**) pressure coefficient distribution with Hill’s spherical vortex model in steady flow. The velocity distribution is shown using streamlines, with intermediate arrow markers to indicate the flow direction.

**Figure 4 bioengineering-12-00279-f004:**
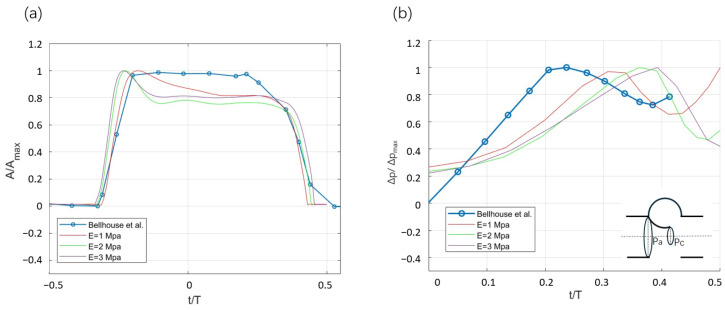
Comparison of experimental results of Bellhouse et al. (1969) [5] and numerical results for cases of different stiffness in (**a**) temporal evolution of valve opening area and (**b**) pressure differences during valve closure where $\Delta p=p_{c}-p_{a}$.

**Figure 5 bioengineering-12-00279-f005:**
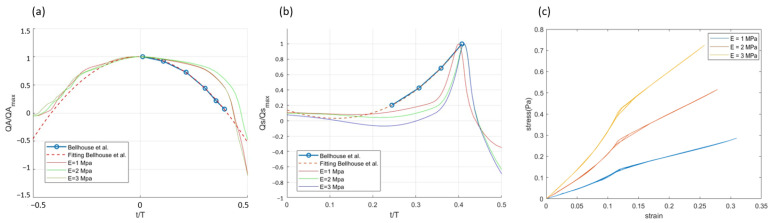
Comparison of experimental results of Bellhouse et al. (1969) [5] and numerical results for cases of different leaflet stiffness in (**a**) temporal evolution of flow into the aorta, (**b**) flow into the cusp–sinus cavity at closure phase, and (**c**) stress–strain curves for different leaflet stiffness.

**Figure 6 bioengineering-12-00279-f006:**
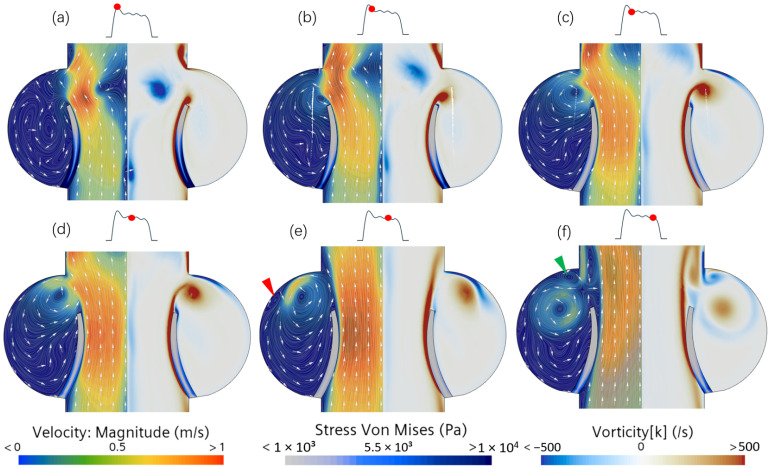
Velocity and vorticity distributions for (**a**–**c**) pre-peak systole, (**d**,**e**) peak systole, and (**f**) post-peak systole. Secondary vortical structures forming after the development of the primary vortex are indicated with red and green arrows. The small inset plot (with the black curve) shows the aortic opening area during the systolic phase, with the red dot indicating the instantaneous time.

**Figure 7 bioengineering-12-00279-f007:**
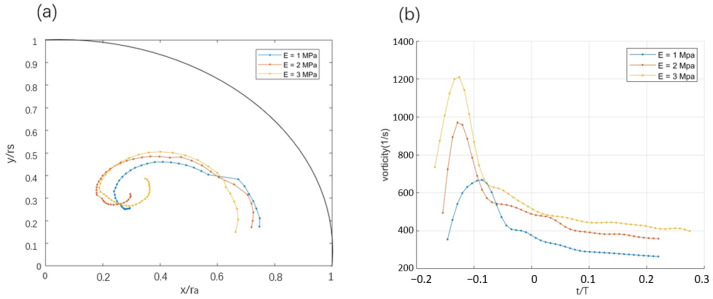
(**a**) The trajectories of the main vortex derived from three stiffness models; (**b**) evolution of the maximum vorticity of the vortex region with time.

**Table 1 bioengineering-12-00279-t001:** Parameters of the 2D aortic model.

Parameter	Notation	Value
Annular radius	$r_{a}$	0.01 m
Sinus radius	$r_{s}$	0.01 m
Upper sinus height	$h_{1}$	0.015 m
Lower sinus height	$h_{2}$	0.005 m
Leaflet tip thickness	$T_{t}$	0.0007 m
Leaflet root thickness	$T_{r}$	0.002 m

## Data Availability

Data available on request due to restrictions in repository access.

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
