# Peer review of "Vortex Dynamics in the Sinus of Valsalva"

_bioengineering, 2025, doi:10.3390/bioengineering12030279_

Round 1

Reviewer 1 Report

Comments and Suggestions for Authors

The manuscript submitted by Fan and Sandrom titled: "Vortex dynamics in the sinus of Valsalva" is an interesting work studying the fluid mechanics phenomena for a potential medical application of an aortic valve.

The paper is well structured and organized. Flows well and the write up is of high quality. The work is nicely and accurately presented.

The reviewer would like to offer a few points for improving an already strong paper:

  1. The authors consider the limitation of blood simulation in their discussion as a brief mention. It would be interesting to theorize how that may change their results and draw from the literature to see if this issue has been addressed by the scientific community.
  2. Consider discussing how the results may be informed by the difference in viscosity (ie Reynolds number), even under the assumption of a "homogeneous blood".
  3. How would vasodilation affect the model presented? 

Reviewer 2 Report

Comments and Suggestions for Authors

The reviewer expresses appreciation for the opportunity to review this manuscript.  The authors have conducted an appropriately designed simulation model studying the fluid dynamics of the aortic root, as it relates to valve leaflet stiffness.  The assumptions of the simulated environment are reasonable given the constraints of such experiments.
The reviewer applauds the succinct yet descriptive nature of the manuscript narrative.
One recommendation for the authors is to provide in the Introduction and Discussion sections, a brief clinical relevance for the study model.  For example, valve stiffness may be relevant to the progressive nature of aortic valve stenosis, and how these predictions may have an impact.  Many readers likely are to draw their own conclusions about the clinical applications.  However, having the authors' own synopsis on its clinical importance of their efforts would improve the overall impact.

Round 2

Reviewer 1 Report

Comments and Suggestions for Authors

The authors have made a reasonable effort in addressing reviewer's comments. Proofreading is suggested.